# Green Innovation and Business Sustainability: New Evidence from Energy Intensive Industry in China

**DOI:** 10.3390/ijerph17217826

**Published:** 2020-10-26

**Authors:** Liang Li, Hajar Msaad, Huaping Sun, Mei Xuen Tan, Yeqing Lu, Antonio K.W. Lau

**Affiliations:** 1School of Business, Nanjing University of Information Science and Technology, Nanjing 210044, China; llcwllcw@hotmail.com (L.L.); msaadhajar@gmail.com (H.M.); mathevas93@hotmail.com (M.X.T.); 2Research Center for Prospering Jiangsu Province with Talents, Nanjing University of Information Science and Technology, Nanjing 210044, China; 3Research Institute for Environmental and Health Nanjing University of Information Science and Technology, Nanjing 210044, China; 4School of Finance and Economics, Jiangsu University, Zhenjiang 212013, China; 5School of Management, Kyung Hee University, Seoul 02447, Korea; yeqing11@naver.com

**Keywords:** green innovation, business sustainability, energy intensive industry

## Abstract

Chinese manufacturing has recently undertaken the responsibility of energy conservation and emission reduction to address climate change. This research analyzes green innovation on business sustainability in the energy-intensive industry in China from the manager perspective, researched data from 229 Chinese managers via structural equation modeling (SEM). The results demonstrated that green innovation had three dimensions: green product innovation, recycling, and green publicity. Business sustainability also had three dimensions: financial performance, environmental performance, and social performance. It also shows that green innovation had a significant effect on business sustainability in the energy-intensive industry. More specifically, we found that recycling has more impact on social performance when compared with green publicity. However, green publicity has a large effect on environmental performance; moreover, green product innovation has more impact on financial performance than green publicity. We also found that environmental performance has a positive effect on financial and social performance results. The alternative models were used to examine the second-order factors of green innovation and business sustainability to test the study’s robustness and supported our findings. Thus, this study contributes to the field by helping managers to make decisions when dealing with sustainable environmental management. It provides new empirical evidence to support the development of a low-carbon circular economy and realization of a carbon-neutral goal by 2060 in China.

## 1. Introduction

Global warming has intensified in recent years, and the increase in extreme weather events worldwide has damaged the ecosystems. Therefore, reducing greenhouse gas emissions and coping with global climate change has become a standard task facing humanity [1,2]. In China, industrial manufacturing and energy-intensive industries still account for a larger share of the economy. This industry can cause an increase in carbon dioxide emissions and severe air pollution. Even the global climate system poses a severe threat that somewhat ignores environmental innovation [3,4]. In this context, more developed countries are engaged in developing energy conservation and emission reduction policies. Its focus on transforming traditional high-carbon industries implies that it would impact corporate performance significantly would harm energy-intensive corporates’ sustainability performance [5].

Extant literature shows that technology innovation can inhibit environmental pollution and ecological damage, and it can also promote firm performance and green development [6,7]. Green innovation includes two components, namely green product innovation and green process innovation [8]. Previous research has indicated that environmental regulations and internal corporate governance would influence green innovation, and green innovation practices can bring many competitive advantages and better business sustainability [9,10]. However, there is a preliminary empirical study on the effect of green innovation among Chinese energy-intensive companies [11]. Many research studies recently indicated that the variable of business sustainability performance was used as a dependent research variable in different countries [12,13,14]. These findings suggest many researchers have been interested in exploring the influencing factors on business sustainability. However, these are relatively insufficient in terms of business sustainability among Chinese energy-intensive industries. For example, a representative study was implemented by Wei et al. The work examined the relationship between environmental management and firm size; an effect on financial performance was not yet observable [14]. Singh et al. found that green innovation would positively correlate with the environmental performance [15].

Eikelenboom and Jong investigated the internal and external integrative dynamic capabilities on environmental and social performance. Their study’s outcomes revealed that external integrative dynamic capabilities had a more considerable positive impact on sustainability performance compared to internal integrative capabilities in small and medium-sized corporations [12]. Pham and Kim have extended the concept of sustainability performance in construction firms [16]. Mousa and Othman explored sustainable performance in healthcare service organizations, suggesting that human resource management practices positively affected business sustainability [17]. However, research on the impact of green innovation practices on energy-intensive companies is limited [18].

This study bridges this gap by investigating green innovation on business sustainability among China’s energy-intensive companies [1,19]. Green innovation is regarded as one of the main trends of sustainable development globally, including such green innovation practices as materials reduction and air pollution prevention [20]. Green innovation allows to address these environmental concerns. Furthermore, it also creates competitive advantages and increases employee happiness [21,22,23]. Consequently, this study aims to test the relationship between green innovation practices and business sustainability among Chinese energy-intensive companies. This study’s findings may provide valuable insights into the effects of green innovation practices on energy-intensive companies. These findings will be useful for the management of green innovation and how it can improve energy-intensive company practices that address climate change, improve business sustainability, and achieve a successful transformation.

This paper is organized as follows. In Section 2, the representative concepts are conducted in the research background. Section 3 provides the research hypothesis and research model. Section 4 describes the research methodology, which includes data collection and analysis and empirical results. In Section 5, we present the conclusions and research discussion, as well as the limitations and further research.

## 2. Research Background

### 2.1. Green Innovation

With the worsening of environmental pollution and global warming, the deterioration of climate change characterized by the increasing carbon emissions, or the greenhouse effect, the sustainable global economy is seriously constrained [24,25]. The emission of carbon dioxide or greenhouse gas has been a significant concern in the world’s economic development [10,26]. Many governments have set environmental regulations to promote environmental technology innovation and reduce pollutant emissions to protect the environment [27,28,29]. China has the world’s largest population, the world’s most extensive product manufacturing, and is the largest emitter of carbon emissions, promoting green innovation in China is crucial [30]. Thus, China is facing enormous pressure to reduce emissions. As the market’s micro-main body, the industry and corporates must adapt to climate change [19,31,32,33].

According to Xie et al. and Dangelico and Pujari’s research, Chinese manufacturing is a primary concern with green product innovation [20,34]. Green innovation practices can be defined as “energy minimization, materials reduction, and pollution prevention during the whole environmental production process with positive sustainable or environmental attributes products” [35]. Green product innovation saves resources and realizes the recycling of raw materials and wastes, avoids wastes, or reduces environmental pollution [36]. According to Dangelico and Pontrandolfo, the term green product innovation was first revealed in the early 1990s [35]. Potter proposed to term it as green product or eco-product early in the 1990s; green product innovation practices consist of business activities that intend to satisfy market needs, as well as diminish the negative impacts on the natural environment [37]. Previous studies suggested that corporate decision-makers and new product development managers have to be concerned about climate change issues [38,39,40]. Roy proposed that corporates operate less wasteful manufacturing processes and develop greener products such as bicycles, greener cars, low energy housing, soft energy lighting, greener washing machines, or others to improve business sustainability [38]. Moreover, Smith et al. suggested that green product innovation is often associated with recycling, reducing waste output, using few materials, and reusing [39].

Dangelico and Pontrandolfo, and Ozaki stated that the green product innovation activity is the innovation activity that focuses on the environment; it incorporates green business activities that consist of green packaging, renewable production process, and publicity [35,41]. Green innovation can be described as the fundamental business principle that highlights environmental ethics. It can help the corporation to improve its reputation with green investment and green development [20,38,41,42]. In general, green innovation practices’ main objectives are to minimize environmental hazards resulting from industrial manufacturing, strengthen the corporate competitive advantage, have significant ecological benefits, and promote the ecological economy [21,35,38,43,44]. Based on previous research, green innovation is corporate attempts to develop eco-social products associated with recycling, use fewer materials, and promote them to the green market. Thus, green innovation can be classified into three elements: green product innovation, recycling, and green publicity.

According to Roy and Smith et al., green product innovation can be defined as the kind of product that is eco-designed. Green innovation within the production cycle is improving in alignment with the market needs and has a significant beneficial effect on the environment [38,39]. According to Dangelico and Pontrandolfo, green products are conducted to minimize the environmental impacts, avoid waste, maximize resource efficiency and reduce the use of harmful chemicals to protect or promote a healthier environment [35]. In the same way, Hazen et al. proposed that the green or eco-product is made to reduce the environmental damage implicated within materials, energy, and pollution. It could involve eco-materials recyclable materials [45]. Moon et al. suggested that green product innovation can be described as environmental or eco-destined to minimize the negative impact of the production cycle on the environment [46]. To conclude, green products can be described as environmental helpers, the kind of products that positively affect sustainable environmental development, and raise environmental awareness that has introduced environmentally-responsible practices or corporate social responsibility in the firms across some industries [18]. Li et al. and Chan studied the link between green publicity and green building and found that green publicity would significantly impact the adoption of green innovation performance [47,48]. Wong has found out that the details mentioned on green product innovation would enhance enterprise competitiveness and reputation [3].

### 2.2. Business Sustainability

Business sustainability is termed as one of the core components of corporate sustainability management. Green human resource management activity can affect the dimensions of environmental and economic performance [15]. Business sustainability can be considered as achieving the environmental goal while meeting sustainable corporate development. Russo and Fouts, and Sharma pointed out the fundamental and essential concept of sustainable performance to organizations. This performance could be seen in environmental pollution governance, social and market share, and profitability owned by the corporates [49,50]. Székely and Knirsch have defined business sustainability as a selection of sustainability connected to social and corporate, increasing society and sustainable economic development [51]. Singh et al. and Mousa and Othman introduced the business sustainability framework. Their framework included the three main elements of sustainability performance, and sustainability measures have been investigated with green innovation [15,17].

Following the introduction of the concept of sustainability, the governments of some countries enforced stringent environmental rules and set sustainable goals to mitigate climate change issues; more companies have also adopted and developed green or environmental innovation. Researchers are interested in investigating environmental innovation’s effects on business sustainability, such as financial performance, environmental performance, and social performance [15,52]. This is consistent with Asadi et al. measures of sustainability from the economic or financial, environmental, and social performance [53]. Singh et al. measured the business’s economic and environmental impacts [15].

Montes et al. termed “financial performance” as the benefits gained during a certain period of operation. It refers to the production and operation achievements, such as the profitability, operation ability, and development ability of an enterprise [54]. Profitability is reflected in the market share of products, sales of new products, profitability, operation, and development ability, reflecting the current value, and future potential of an enterprise in its survival and development. Wang and Berens proposed that corporate social responsibility played a vital role in financial performance. Under the green innovation-based social responsibility, this concept is also considered critical and essential, correlated directly with innovation performance [55]. According to Javed et al. research findings, who suggested that corporate social responsibility can play a role in improving the firm financial performance, responsible leadership could encourage corporate reputation and performance [56].

Environmental performance is termed as one of the core components of business sustainability. It can also be defined as enterprises’ performance to reduce carbon emissions, such as environmental incident reduction, renewable materials increase, waste reduction, and resource consumption [52]. The researchers suggested that the next phase of the corporate sustainability manager should promote green transformation, reduce carbon emissions, and promote social and environmental and economic sustainable development [57]. Long et al. and Lee and Min argue that corporate green innovation has an active role in environmental performance among manufacturing [6,52]. Some researchers explained that social performance is influenced by corporate reputation and the firm’s size, corporate governance [58,59].

Social performance is the concept extended from corporate social responsibility and its emphasis on actual results achieved. It emphasizes the responsibilities of economic stakeholders [60]. Many researchers have argued that corporate social performance is a broad concept used to describe a wide range of diverse phenomena. Corporate social responsibility is ineffective if it does not lead to corporate social performance. Social performance is related and extended from the corporate social responsibility initiative. Still, it differs as it focuses more on the result and whether a corporate social responsibility project delivers expected results or outcomes [61]. Swanson also suggests that personal values and ethics, such as the executive’s sense of morality, significantly affect corporate social performance [62]. Corporate social responsibility is positively related to financial performance; it also leads to green innovation and social performance in the manufacturing industry [63,64].

This paper aims to investigate that the factors that affect energy-intensive corporate business sustainability from the perspective of three aspects, such as financial performance, environmental performance, and social performance set of associations by energy-intensive corporates that allow sustainability to gain strength. However, there is a lack of studies of energy-intensive companies’ business sustainability. Therefore, the major elements are financial performance, environmental performance, and social performance, forming a relevant base for business sustainability in China in the energy-intensive industry.

## 3. Research Hypothesis and Theoretical Model

### 3.1. Research Hypothesis Development

According to Schumpeter’s theory of innovation, innovation is new ideas for how things are done. Innovation is a driving force of corporate competitive advantage, economic growth, and societal and technological progress. Based on disruptive innovation theory, corporate’s innovation is a driving force of corporate performance.

The influence of a growing demand for environmental products has led to sustainability becoming an integral part of business across all industries [20,65]. Previous studies conducted by Wong suggested a significant effect of green innovation on company competitiveness [3]. An increasing number of Chinese firms are green product-certified based on ISO14001, which reflects companies’ growing interest in green product innovation and environmental management. This gives an overview of how China can continue developing its social-economic activity [66,67]. According to Xie et al., both green process innovation and green product innovation could improve corporate financial performance. The results also suggested that green product innovation mediates the relationship between green process innovation and corporates financial performance [36]. The following hypotheses on green innovation practices may have a positive influence on financial performance, which is drawn from the literature:

**Hypothesis** **1 (H1).**
*Green innovations practice elements (a) green product innovation, (b) recycling, (c) green publicity have a positive influence on financial performance.*


Technological innovation produces many positive external effects, such as reducing pollution, but it can also have positive social impacts; environmental innovation particularly is increasingly used as an opportunity to promote corporate contribution to sustainable development [38]. The environmental performance involves the use of energy and resources, as well as emissions and waste. Recycling is the method or process of collecting waste items and converting them into new products and reusable materials [38,39]. This process actively contributes to sustainable development and may save natural resources by using waste as raw material [68,69]. Xie et al. stated that green innovation practices significantly affected company’s environmental performance and competitive advantage [34]. Especially, green innovation explained changes in environmental performance. Green environmental methods, such as green design, ecological design, and green production planning are increasingly popular [21]. Generally, “green products” are defined as products that minimize their environmental footprint throughout their life cycle. The production of green products involves eliminating toxic substances and the prevention of waste [68].

Recycling avoids tapping into natural deposits such as timber, water, and minerals and helps limit the amount of waste and remains to be buried, and reduce pollution by minimizing new raw materials collection and reducing energy consumption [69,70]. Several studies have suggested that some companies use recycling as an everyday activity that may provide financial savings and federate employees around common eco-values. Corporate recycling behavior is a component of green innovation practices that can enhance company advantage [6,39,65,69]. Green publicity refers to the publicity or social media report type that promotes eco-friendly operations, sustainable policies, and green measures concerning the activities of the company [71]. Green consumers view green publicity as an effort to assess the effect of publicity and its benefit to the environment [72]. Firms use social media or news to communicate at persuading consumers of the environmental services and positive publicity of green products to inhibit pollution. Green publicity can enhance the environmental performance and social performance of the corporates [71,73].

Green innovation can significantly reduce their costs and have the opportunity to improve social performance, and recycling practices positively impact the performance of companies [74]. Previous studies conducted by Gerrard and Kandlikar and Chen et al. have suggested a significant effect of green products, recycling efforts, and environmental policy management on corporate performance [75,76]. Moreover, the growing demand for green products has led to sustainability becoming an integral part of business across all industries [77]. Amores-Salvadό et al. stated that implementing environmental innovation practices to introduce it to consumers, who would have a more positive significant effect on consumers’ response to the products, environmental product innovation may positively influence corporate reputation, which can enhance firm performance [78]. Green product innovation has opened new opportunities for firms, and environmental products have given them the possibility of innovation and a positive active reputation [79]. Lin and Niu pointed out that green innovation practices had a positive effect on corporates reputation. It helps develop the corporate-awareness in the consumer. It allows them to see the contributions of firms in raising environmental awareness and helping to solve some of the ecological issues [73]. The following hypotheses on green innovation have a positive influence on environmental performance, which is drawn from the literature:

**Hypothesis** **2 (H2).**
*Green innovations practice elements (a) green product innovation, (b) recycling, (c) green publicity have a positive influence on environmental performance.*


Corporates’ action to share information about green innovation via channels such as local newspapers, company websites, and social media accounts can be considered green publicity. Green publicity can improve information transparency and strengthen the management of environmental violations, thus improving corporate environmental performance. The higher the green publicity as information disclosure, the lower the information uncertainty and asymmetry of green technology innovation; it can also enhance institutional investors’ recognition of green technology innovation; thus, improving corporate performance and environmental performance. Based on the theory of corporate reputation, environmental violations of energy-intensive enterprises directly affect reputation. Corporate reputation will reduce corporate performance; the reputations of corporate managers will also be affected. Thus, based on the reputation and corporate social responsibility, green publicity of environmental innovation can enhance corporate reputation and social performance [55]. Green publicity can also lead to other’s corporate environmental awareness that can create environmental innovation by providing information and knowledge, influence their environmental behavior leading to a positive social and environmental performance. Based on corporate social responsibility, green publicity can help corporate create a positive reputation, enhance environmental performance and social performance, and indirectly contribute to increment in financial performance [56].

Moreover, energy-intensive corporates can change their operating and production models, adopt green technology innovation, reduce the additional costs associated with carbon trading and taxes, and use energy-intensive enterprises to treat more pollution and carbon emissions. The green innovation technology will reduce enterprises’ total cost and improve environmental performance, thus improving financial performance and social performance. The study of Shaukat et al. determined the relationship between corporate social responsibility strategy with environmental and social performance. The findings indicated a significant positive relationship between green market orientation with the environmental and social performance [80]. The following hypothesis on green innovation practices may have a positive influence on social performance and is drawn from the literature:

**Hypothesis** **3 (H3).**
*Green innovation practice elements (a) green product innovation, (b) recycling, (c) green publicity have a positive influence on social performance.*


### 3.2. Theoretical Model

According to the above discussion, the research model is proposed as following: we investigate the effects of green innovation on business sustainability, we analyze the influence of green product innovation (GP), recycling (RE), green publicity (GP), financial performance (FP), environmental performance (EP), and social performance (SP). As shown in Figure 1.

## 4. Methodology and Results

### 4.1. Survey Design and Measurement

The questionnaire designed for this research includes two sections. In the Section 1, the sample’s characteristics are discussed: gender, age, education, income, and types of industries. As for the Section 2, the respondents were asked to read statements and determine their degree of agreement using a 5-point Likert scale. The questionnaire items used in this study were developed and adapted based on the existing literature. More specifically, the questionnaire surveyed green innovation practices with three components: recycling, green product innovation, green publicity, covered by three items, each from Roy and Dangelico and Pujari. Recycling was measured by using three-item versions of the scales [20,38]. Example items are “our corporate makes efforts to use recycled materials”, “our corporate uses reusable materials”, “our corporate uses recycled or reusable materials in packaging.” Green product innovation was measured by using three-item versions of the scales. Example items for green product innovation included “our corporate uses environmental considerations in product design”, “our corporate uses ecological and green materials in manufacturing”, “our corporate invests in R&D programs to create environmentally friendly products.” Green publicity was measured by using a three-item version of the scales; example items for green publicity included “our corporate prefers digital communication tool for promoting eco-friendly products”, “our corporate publicity about this green product innovation”, “our corporate promotes the new green ideas to the consumers.”

Business sustainability included three components: financial performance, environmental performance, and social performance [12,13,51]. Business sustainability was measured using the scales developed by Yusliza et al. and Eikelenboom and De Jong and Mousa and Othman. Financial performance was measured by using four items of the scales. For example, “in the last three years, this company’s sales performance was increased”, “return on investment was increased”, “The competitive market share was expanded”, “mainly business revenues were increased”. Environmental performance was measured by using four items of the scales. “This corporate increases recyclable, reusable, and recyclable material use”, “this corporate reduces carbon dioxide emissions”, “this company reduces the discharge of waste-water and chemical pollutants”, “this corporate reduces the consumption of dangerous/harmful/toxic/substances.” Social performance was measured by using three items of the scales, for example: “this corporate increases activity to protect the natural environment”, “this corporate increases the social reputation”, “this corporate develops community economic activities and provide more employment opportunities”.

### 4.2. Sampling

According to the above discussion, this paper chooses the energy-intensive industry as research evidence because China is in a period of low carbon transition. The supply-side reform of green technology innovation is fundamental in the energy-intensive industry. However, there is a lack of literature on green innovation on business sustainability among energy-intensive corporates. There are many enterprises in the energy-intensive sector in China. Besides, according to the database of The State Intellectual Property Office of China, environmental patents are mainly concentrated in energy-intensive enterprises. Thus, the energy-intensive industry is used as evidence in this present study.

A Likert 7-point scale is used, and the degree from 1 to 7 represents degrees of agreement from low to high (Table 1). The survey has been done from the 8th of January to 28th of November, 2019, and 8th of May to 10th of July 2020. Before handing out the questionnaire, we did the pretest for checking the word sentence. We distributed around 20 questionnaires to the manager working at energy-intensive companies in Nanjing industrial parks and Suzhou industrial park. The city of Nanjing is recognized as one of the four great ancient capitals of China. There are more than five industrial parks in Nanjing. It attracts a large number of companies every year.

Suzhou is one of the most important and critical central cities in Jiangsu province. The famous Suzhou Industrial Park is located there. All of them can understand the question items. After this process, we distributed questionnaires to five managers who work in different industries, and finally, we collected 120 questionnaires offline. We also distributed online questionnaires to managers who work at thermal power, iron, steel, building, coal, the automotive, pharmaceutical, and chemical fertilizer industries in China. We distributed a total of 380 questionnaires, but we only collected 109 questionnaires online. The sampled managers from Shanxi, Tianjin, Shandong, and Liaoning answered the questionnaire and got random 10 Yuan RMB rewards. The participation was voluntary.

A total of 500 questionnaires were distributed, leaving 229 questionnaires for the analysis. Table 1 shows the respondents’ demographic characteristics, where 118 (51.5%) were males, 111 (48.5%) were females. Seventy-two (31.5%) were at the age of 21 to 30; 77 (33.6%) was at the age of 31 to 40; 80 (34.9%) were at the age of 41 to 50. In terms of work experience, 97 (42.4%) had 12–36 months, 104 (45.4%) 37–72 months, 28 (12.2%) over 73 months of experience. Lastly, 33 (14.4%) were from thermal power, 36 (15.7%) were from iron and steel, 27 (11.8%) were from building and construction, 25 (10.9%) were from coal and mining, oil or energy industry, 71 (31%) and 37 (16.2%) were from the automotive and pharmaceutical industry. Based on the results, the statistical data related to the respondents are consistent with Mousa and Othman’s (2020) research, which suggests our sampling is representative [17].

### 4.3. Confirmatory Factor and Reliability Analyses

The average variance extracted (AVE) and composite reliabilities (CR) were conducted by PLS 2.0. The results of validity and reliability analysis are presented in Table 2. The average variance extracted (AVE) of green product innovation is 0.678, and recycling is 0.753, and green publicity is 0.780, financial performance is 0.612, environmental performance is 0.624, and social performance is 0.677. The composite reliability (CR) of green product innovation is 0.863, and recycling is 0.901, green publicity is 0.913, and financial performance is 0.862, and environmental performance is 0.868, social performance is 0.863.

The reliability analysis and factor analysis were conducted by PLS 2.0. the results are also shown in Table 2. The Cronbach’s alpha for green publicity is the highest one compared to other latent variables. Recycling is 0.836, green product innovation is 0.762, financial performance is 0.785, environmental performance is 0.795, and social performance is 0.763. Through the reliability test, the AVE and the corresponding CR values are more significant than 0.5 and 0.7, respectively. Besides, all of the Cronbach’s alpha values are greater than 0.7, which means that reliability is verified.

### 4.4. Correlations and Convergent Validity Analyses

The correlation analysis results demonstrate that the relationship between green product innovation and financial performance is positive and significant; the correlation coefficient is 0.643. It means that there are strong positive relations. Similarly, there is a positive relation between green product innovation and environmental performance, and the correlation coefficient is 0.553, as well as between green product innovation and social performance (0.584), recycling and financial performance (0.704), recycling and environmental performance (0.753), recycling and social performance (0.676), green publicity and financial performance (0.630), green publicity and environmental performance (0.639), green publicity and social performance (0.550). The correlation test results show that most of the coefficients are more significant than 0.4 and less than 0.9. Although the correlation coefficient is over 0.6, but, the variables of squared roots of the average variance extracted (AVE) are more significant greater than the correlation weights of each variable. On this basis, we can consider all the variables of discriminant validity as acceptable (See Table 3).

### 4.5. Hypothesis Testing by Using Structural Equation Modeling (SEM) Analysis

Many researchers have conducted the structural equation modeling (SEM) analysis in the corporate sustainability management research field. Preziosi et al. showed the SME analysis to explore consumer perception with corporate environmental practices [81], and Long et al. examined the antecedents of environmental performance [52]. Moreover, Asadi et al. have explored how green innovation activity would influence service sustainability performance using the SEM model. In the present research, we extend and use structural equation modeling analysis by PLS 2.0. to estimate the causal relationship between each variable in the energy-intensive industry [53].

Green product innovation has a significant positive relationship with financial performance and environmental performance (b = 0.356, *t*-value = 4.243; b = 0.190, *t*-value = 2.009), thus, the Hypotheses 1a and 2a were accepted, which supports Dangelico et al. and Amores-Salvadó et al. [35,74]. Besides, there is a positive relationship between green product innovation and social performance (b = 0.293, *t*-value = 2.727). Thus, Hypothesis 3a is accepted, which supports Zailani et al. and Singh et al. [15,82].

Moreover, the SEM modeling analysis shows that recycling has a positive relationship with financial performance (b = 0.353, *t*-value = 4.100); thus, H1b was verified. Also, there is a positive relationship between recycling with environmental performance (b = 0.297, *t*-value = 4.806); thus, the Hypothesis 2b is accepted, which supports and Lin et al. [77], there is a positive relationship between recycling and social performance (b = 0.409, *t*-value = 4.067); thus, the Hypothesis 3b is accepted, which supports Roy et al. and Chen et al. [38,76].

Lastly, the green publicity would significantly impact financial performance; thus, Hypothesis 1c is accepted. Also, green publicity would positively impact environmental performance; thus, Hypothesis 2c is supported (b = 0.409, *t*-value = 3.286), supporting Li et al. and Chan et al. [38,48]. However, green publicity would not positively impact social performance (b = 0.205; *t*-value = 1.836); thus, Hypothesis 3c supports Wong [3]. This may be due to the publicity does not produce substantive social performance. We have also conducted structural equation modeling (SEM) analysis to test if the environmental performance influences financial performance and environmental performance. The findings of this research are as following: environmental performance has a positive relationship with social performance and financial performance (b = 0.679, *t*-value = 15.202; b = 0.767, *t*-value = 16.685) (See Table 4).

### 4.6. Hypothesis Testing by Second-Order Factors into the First-Order Factors

To further confirm if the proposed model and the findings are robust, we first used alternative models to explore the first-order factors transformed into second-order factors. Green innovation has three dimensions (green product innovation, recycling, green publicity); business sustainability has three dimensions (financial performance, environmental performance, and social performance). All of their standard loadings are higher than the recommended criteria of 0.7. Moreover, green product innovation has positive effects on business sustainability (b = 0.860, *t*-value = 26.00). We found that green product innovation could facilitate financial performance, environmental performance, and social performance. This provides empirical verification of this research hypothesis, which has suggested a positive effect of green innovation on business sustainability in Chinese energy-intensive industries (See Figure 2).

## 5. Conclusions

### 5.1. Discussion and Theoretical Contribution

Following the carbon-neutral goal, more Chinese companies are going to practice corporate environmental responsible practice, and more entrepreneurs pay attention to global warming [1,2,83,84,85]. Thus, with increasing attention on corporate practices with green innovation for dealing with climate change [38,86,87], research interest has also been stimulated on measuring the relationship between green innovation practices and business sustainability performance [22,87,88], and environmental benefits and promote low carbon and circular economy [35,38,43,89]. Moreover, green innovation and firm performance [77] and green innovation with energy efficiency [30]. Previous studies analyzed green innovation and environmental performance [52]. In previous findings, the energy-intensive industry is one of China’s main drivers of carbon emissions, which is the main challenge for green transformation. This implies that it would help with energy conservation and emission reduction by Chinese manufacturing, significantly reducing their carbon dioxide emissions and achieving a carbon-neutral goal in 2060.

This research aimed to explore how the green innovation of Chinese manufacturing may affect business sustainability. According to the empirical findings, this present study proposes a three- dimensional aspect of green innovation. It is composed of three distinct components, including green product innovation, recycling, and green publicity. We have identified the effect of green innovation on business sustainability. The finding of this research is as follows: first, green product innovation had significantly affected financial performance, environmental performance, and social performance. Second, recycling had significantly affected financial performance, environmental performance, and social performance in the energy-intensive industry. Third, green publicity also had significantly affected corporates financial performance and environmental performance. Furthermore, we also have found that environmental performance can enhance financial performance and social performance. We have identified that green innovation can increase business sustainability in Chinese energy-intensive companies. These findings support previous studies [3,20,55,65].

Based on these findings, this study can provide some theoretical and practical implications. The policy recommendations for promoting the low carbon transformation of energy-intensive industries in China are put forward as follows. Specifically, it includes the following two aspects: first, from the perspective of green technology innovation, promote sustainable development in different types of energy-intensive industries, and achieve high-quality development of China’s economy. Industrialization and economic growth have brought massive energy consumption and environmental pollution, primarily due to the continuous sustainable development and economic growth of energy-intensive severe challenges to the sustainable development of China’s society. How to deal with this challenge has attracted extensive attention from the government policy-makers and researchers worldwide. According to previous studies, green innovation can save energy resources, reduce carbon emissions and environmental pollution, and waste recycling low-carbon environmental innovation can reduce raw materials. Based on these findings, this study can provide some theoretical and practical implications.

The policy recommendations for promoting the low-carbon transformation of energy-intensive industries in China are put forward as follows. Specifically, it includes the following two aspects: first, from the perspective of green technology innovation, promote sustainable development in a different type of energy-intensive industries, and achieve high-quality development of China’s economy. Industrialization and economic growth have brought massive energy consumption and environmental pollution, primarily due to the continuous sustainable development and economic growth of energy-intensive severe challenges to the sustainable development of China’s society. How to deal with this challenge has attracted extensive attention from the government policy-makers and researchers worldwide. According to previous studies, green innovation can save energy resources, reduce carbon emissions and environmental pollution, and waste recycling low-carbon environmental innovation can reduce raw materials.

Moreover, it can solve the excessive energy resource consumption and control environmental severe pollution to provide an important alternative path. A critical empirical study indicates that the components of green innovation are constructed using a secondary confirmatory factor analysis method. It is better to enhance the level of green product innovation and recycling, also it is better to include even the green publicity of eco-production, which can reduce corporates’ carbon emission in China, in turn, which can help build the socio-economic and environmentally sustainable development goals [27]. Thus, we suggest the Chinese government based on the energy intensive, such as thermal power plant, energy-intensive, iron and steel, building materials and construction, chemical and pharmaceutical, the auto industry puts forward to promote low-carbon green technological innovation incentive policy, to promote the transformation of polluting energy-intensive enterprise green. For example, for the traditional industrial areas, such as the northeast old industrial base, to promote the green transformation of coal mines in Liaoning, and establish traditional enterprises for the conversion of low carbon and operation of green coal chemical industry with coal-to-oil and fine chemical products. Artificial intelligence blockchain and other technological innovations promote mechanized coal mining. It also vigorously develops renewable energy technology to provide energy security for the low-carbon economy’s high-quality development to achieve the carbon-neutral target.

According to the research findings, e-waste is the critical factor of global climate change and air pollution. In China, e-waste, such as refrigerators, mobile phones, and automobiles can create massive pollutant emissions and affect residents’ health; it poses a severe challenge to address global warming. With the development of technology and industry 4.0, it is also crucial to guide green technology innovation in the electronics industry. To offer welfare subsidy policy, it leads enterprises to green low carbon production, implements low-carbon industrial modernization, and establishes green technology innovation energy-intensive the enterprises’ supervision system. Also, the building industry is contributed to high carbon emission with the development of China’s new-type urbanization. We also suggest that developing green innovation through low carbon buildings is vital in the future. It is better to establish the building industry’s regulatory system for green technology innovation. It is better to promote low carbon technology building. It is better to encourage photovoltaic and energy-saving technology in the buildings industry and encourage recycling used steel to reuse and remanufacture. Moreover, we suggest thermal power and energy enterprises to develop renewable energy generation, to reduce fossil energy use, while supporting industrial and residential electricity use, to ensure that the Paris Agreement is achieved while promoting a low-carbon economy in China.

Second, according to the Chinese 2060 years carbon neutral policy, we also recommend that it be better to develop circular business model innovation. It also is better to develop renewable energy technology and low carbon cycle industrial chain. It is necessary to promote the publicity of Chinese green innovation practices through information disclosure, because through newspaper and social media reports, it can improve the transparency of information, to enhance supervision effect, to constraint on corporates’ environmental violations, and thus promote green innovation, which can promote Chinese manufacturing industry to green and low carbon transformation, and promote the Chinese manufacturing’s green and sustainable development [41]. The results of this study may be used as fundamental data in the strategies of green innovation. Particularly. investigating the promoting the effects of green innovation to business sustainability in Chinese energy-intensive companies, which can also help company policymaker as they work dealing with low carbon policy and sustainable development [1,2,38]. Moreover, to promote green technology innovation from green finance, we suggest that the government completed green credit and green bonds and other related carbon trading or related green financial policy.

### 5.2. Limitation and Further Research

This present study explored how green innovation promotes business sustainability among Chinese energy-intensive companies. Moreover, we found the robustness of green innovation on Chinese manufacturing’s business sustainability by using the first-order factors of the structural equation model. However, this research has some limitations.

Firstly, as we collected the data from managers in China’s energy-intensive industry, it is possible that the respondents may not have been fully knowledgeable about all the corporates green innovation and performance, because the respondents were from different departments. Some respondents were not even from the R&D department; thus, we recommend further researchers to deliver to randomly unknown respondents from the R&D department online or offline. We recommend future researchers to collect secondary data from listed companies for empirical analysis in the energy-intensive industry; because the cross-sectional data limited us to examine the casual relationship between green innovation and business sustainability.

Second, in this study, as energy-intensive enterprises with better green innovation are selected for the sampled population, we could not control for the considered physical locations of the different regions among energy-intensive firms. It is possible that different provinces or cities have different green finance policy that affect the environmental performance of the firms. We suggest investigating green innovation needs high quality system as a guarantee, especially in the design of an incentive system; it needs to increase financial support, in particular, the construction of low carbon industrial chain in poor or economically backward areas, to develop the circular business model innovation while improving the employment and quality of life of people living in low and underdevelopment regions, and reflect the organic unity of government, significant market, and capable enterprises. We also recommend that secondary data on green technology innovation of energy-intensive enterprises is collected to discuss government subsidy, information disclosure, green technology innovation, environmental innovation capability, and environmental performance [5,7,12,28,43].

## Figures and Tables

**Figure 1 ijerph-17-07826-f001:**
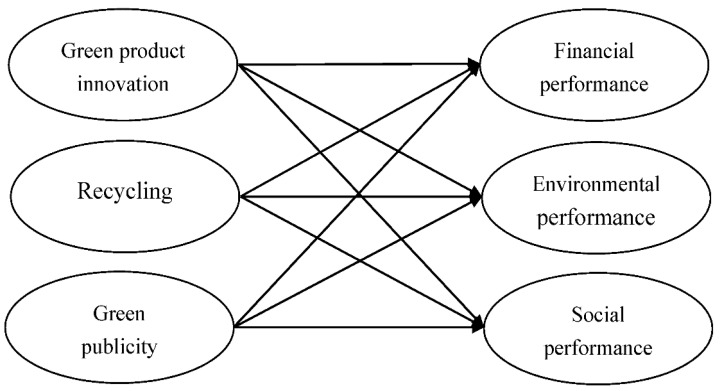
Research model.

**Figure 2 ijerph-17-07826-f002:**
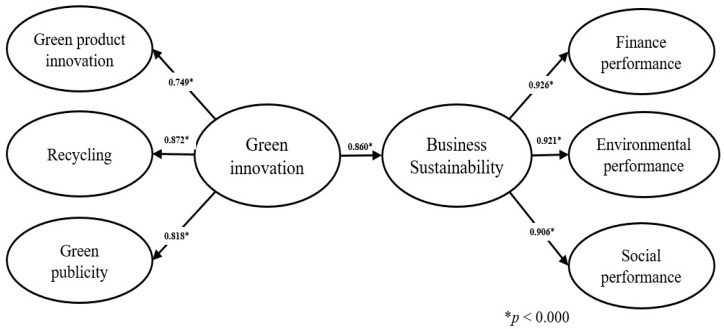
An alternative model for transforming the second-order factors into the first-order factors of green innovation and business sustainability.

**Table 1 ijerph-17-07826-t001:** Demographic characteristics.

Measurements	Types	Numbers	Percentage
Gender	Female	118	51.5
	Male	111	48.5
Age	21s–30s	72	31.5
	31s–40s	77	33.6
	41s–50s	80	34.9
Education	Undergraduate degree or under	135	59
	Master’s degree or over	94	41
Work experience	12–36 months	97	42.4
	37–72 months	104	45.4
	73 months	28	12.2
Type of industry	Thermal power plant	33	14.4
	Iron and steel	36	15.7
	Building and construction	27	11.8
	Mining or oil energy industry	25	10.9
	Auto industry	71	31.0
	Pharmaceutical and chemical fertilizer industry	37	16.2

**Table 2 ijerph-17-07826-t002:** The results of validity and reliability analysis.

Latent Variable	Observable Measurements	Standardized Regression Weight	AVE	C.R.	Cronbach’s α
Green product innovation	Green product innovation1	0.833	0.678	0.863	0.762
	Green product innovation2	0.870			
	Green product innovation3	0.765			
Recycling	Recycling1	0.876	0.753	0.901	0.836
	Recycling2	0.836			
	Recycling3	0.890			
Green publicity	Green publicity 1	0.862	0.780	0.913	0.858
	Green publicity 2	0.881			
	Green publicity 3	0.905			
Financial performance	Financial performance1	0.821	0.612	0.862	0.785
	Financial performance2	0.866			
	Financial performance3	0.783			
	Financial performance4	0.639			
Environmental performance	Environmental performance1	0.779	0.624	0.868	0.795
	Environmental performance2	0.869			
	Environmental performance3	0.830			
	Environmental performance4	0.665			
Social performance	Social performance1	0.817	0.677	0.863	0.763
	Social performance2	0.829			
	Social performance3	0.823			

**Table 3 ijerph-17-07826-t003:** Results of inter-variables correlations and convergent validity.

Variable	GP	RE	GPI	FP	EP	SP
Green product innovation (GP)	**0.873**					
Recycling (RE)	0.509	**0.914**				
Green publicity (GPI)	0.384	0.570	**0.926**			
Financial performance (FP)	0.643	0.704	0.630	**0.886**		
Environmental performance (EP)	0.553	0.753	0.639	0.754	**0.892**	
Social performance (SP)	0.584	0.676	0.550	0.778	0.742	**0.873**

Note: N = 229, squared roots of AVE extracted are shown in boldface on the diagonal and variable correlations are below the diagonal.

**Table 4 ijerph-17-07826-t004:** The results of hypothesis validation.

Hypothesis Path	Path Coefficients	*t*-Value	Testing Results
Green product innovation→Financial performance	0.356	4.243	Accept
Green product innovation→Environmental performance	0.190	2.009	Accept
Green product innovation→Social performance	0.293	2.727	Accept
Recycling→Financial performance	0.353	4.100	Accept
Recycling→Environmental performance	0.297	4.806	Accept
Recycling→Social performance	0.409	4.067	Accept
Green publicity→Financial performance	0.297	3.030	Accept
Green publicity→Environmental performance	0.409	3.286	Accept
Green publicity→Social performance	0.205	1.836	Reject

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
