# Peer review of "Green Innovation and Business Sustainability: New Evidence from Energy Intensive Industry in China"

_ijerph, 2020, doi:10.3390/ijerph17217826_

Round 1

Reviewer 1 Report

The topic is very intereting and relevant. However, the english is poor and sentences are not well writen,

Maybe it would be interesting to do a Table with all the presented definitions and concepts. It would help readers to be more attentive and understand much better all the perspectives.

I am afraid that Hypothesis are not clear explained - they seem all the same, when in fact they try to study different components. I suggest to improve all of them.

The methodology seems to be very interesting and robust.

I suggest you to improve all the text (shorter sentences, better english, more Tables).

I am not in accordance with your statement: ==> 241 - "technological innovation produces many negative impacts...." ==> please eplain this better in your text

I would like to congratulate the authors, but I suggest to improve the text and english.

Author Response

Response to Reviewers

Manuscript number:  ijerph-939554

Title: Green Innovation Practices and Business Sustainability: New Evidence from Energy Intensive Industry

Dear reviewer,

Many thanks for a set of knowledgeable scholars' constructive suggestions on our working manuscript entitled: “Green Innovation Practices and Business Sustainability: New Evidence from Energy Intensive Industry” (Manuscript number: ijerph-939554). Those comments are very helpful for improving our paper, as well as providing the important guiding significance to our research. According to the editor’s and reviewers’ comments, we have made a serious revision on this research. The revisions were marked in red below point by point.

Thank you very much again for your suggestions.

Best regards

Li Liang

Reviewer 2 Report

Thank you. Interesting article.

The topic is vital. The methods and sample size are good, and the conclusions are important and will be interesting to both researchers and managers. I am in support of publishing it, if it is edited by a native speaker to improve the writing.

Some improvements really have to be made. Almost every sentence needs to be slightly revised to be gramatically perfect.

In some cases, the errors make the sentence difficult to understand. The last sentence is not understandable: "Second, in this 469 research we did not consider different industry of green innovation practices of different industry, it
470 is better to consider the adoption of green innovation actvities in different industry (e.g. green IT)."

As another illustrative example: in the abstract, there are misspellings: "More specifically, we have 23 found that recycling has a laerge effect on social performance comapre with green publicity and 24 green product innovation."

Also, I very much recommend citing the following articles for the following propositions:

At line 46, I would add one more consistent and supporting source after footnotes 6 and 7 (as footnote 8) - so that the references appear "[6,7,8]": Wu, J., Liu, L. & Sulkowski, A.J. (2011). Environmental Disclosure, Firm Performance, and Firm Characteristics: An Analysis of S&P 100 Firms. Journal of the Academy of Business and Economics, 10 (4), 73-84.

After Footnote 15 in line 58: "Previous research has examined the relationship between environmental management and firm characteristics in China, and found that, at the time, an effect on financial performance was not yet observable, yet a relationship to firm size was seen." Wei, L., Wang, W., Sulkowski, A. J. & Wu, J. (2011). The Relationships between Environmental Management, Firm Value and Other Firm Attributes: Evidence from Chinese Manufacturing Industry. International Journal of Environment and Sustainable Development, 10 (1), 78-95.

Next, on line 74, after Footnotes 23-25: "In turn, a greener reputation has been found to have a greater impact on worker happiness than financial performance." Walsh, C. & Sulkowski, A.J. (2010). A Greener Company Makes for Happier Employees More So Than Does a More Valuable One: A Regression Analysis of Employee Satisfaction, Perceived Environmental Performance and Firm Financial Value. Interdisciplinary Environmental Review, 11 (4), 274-282.

As far as the methodology and execution and sample size and conclusions: I thought all of that was solid and interesting and valuable.

I might rearrange the final sections so that the Limitations section comes before the Conclusions section. 

Minor typos: 

There are many... but please review everything carefully, all the way to the end - for example - Endnote 30: the number 30 is repeated.

Author Response

Detailed Response to Reviewers

Manuscript number:  ijerph-939554

Title: Green Innovation Practices and Business Sustainability: New Evidence from Energy Intensive Industry

Many thanks for a set of knowledgeable scholars' constructive suggestions on our working manuscript entitled: “Green Innovation Practices and Business Sustainability: New Evidence from Energy Intensive Industry” (Manuscript number: ijerph-939554). Those comments are very helpful for improving our paper, as well as providing the important guiding significance to our research. According to the editor’s and reviewers’ comments, we have made a serious revisions on this research. The revisions were marked in red below point by point.

Thank you very much again for your suggestions.

Best regards

Li Liang

Reviewer 3 Report

The topic of the Paper "Reprintable Paste-Based Materials for Additive 2 Manufacturing in a Circular Economy" is of high interest to the reader as today there is no Circular Economy known in AM . As many AM processes use plastic compounds the amount of waste is increasing. An Approach to reuse manufactured 3D products is of high interest.

The study as some minor drawbacks but is well worth to be published. It should be taken into consideration that the study was performed without any project funding and therefore less resources are available.

I comment on specific issues as following:

Chapter 1 Introduction: is fine and is a good intro to the topic

Chapter 2 Materials and Methods: Here I was missing the circular economy approach of the material selection. Mussel shells are not everywhere available. What was your motivation to select this material? Is this a waste product in your region? What happens if you want to produce tons of 3D figures? Is it feasible to select this material? (This also fits to the conclusion section in the end). The same applies to the alternative matierals as cacao, pine, etc.

Line 145 -156 must be deletec, as it is part of the format style and does not belong here.

Chapter 2.8 is irritating, as it is unclear if Figure 4 also represents the hair pin. Maybe you write in chapter 2.7. that later in the process a prototype was created by .... and this is shown later in figure 8 as result. Otherwise the reader is irritated. It would also be nice to give a scale in figure 4 to imagine the size.

Chapter 4 Line 335. a word is missing here between "the printed....is returned"

In chapter 4 I was missing more discussion on the alternative materials, as the only statement was, that they are comparable to mussel shell paste. Can you be more specific? Is the reversibility process the same with the other materials? What needs to be done? Where are the problems?

Chapter 5: I was missing a focus on the material input here, as the paper is on Circular Economy. Mussel shells are not everywhere available and how do you overcome this issue? This is important as you proposed alternative materials but does not present their results properly . They were only marginal presented in table 4.

Overall , I strongly recommend to publish this paper, as it is of high intereset for the readers and addresses an important topic. As my criticism to the paper is easy to adjust, I recommend minor changes.

Author Response

Detailed Response to Reviewers

Manuscript number:  ijerph-939554

Title: Green Innovation Practices and Business Sustainability: New Evidence from Energy Intensive Industry

Many thanks for a set of knowledgeable scholars' constructive suggestions on our working manuscript entitled: “Green Innovation Practices and Business Sustainability: New Evidence from Energy Intensive Industry” (Manuscript number: ijerph-939554). Those comments are very helpful for improving our paper, as well as providing the important guiding significance to our research. According to the editor’s and reviewers’ comments, we have made a serious revision on this research. The revisions were marked in red below point by point.

Thank you very much again for your suggestions.

Best regards

Li Liang

Round 2

Reviewer 1 Report

Much better explaind and improved to be published.

minor corrections neede (please some suggestions).

Congratulations once again.

Author Response

Many thanks for the knowledgeable scholar's constructive suggestions

on our working manuscript entitled: “Green Innovation Practices and

Business Sustainability: New Evidence from Energy Intensive Industry”

(Manuscript number: ijerph-939554). Those comments are very helpful

for improving our paper, as well as providing the important guiding

significance to our research. According to the editor’s and reviewers’

comments, we have made serious revisions to this research. The revisions

were marked in red below point by point. We have revised it round by round, we have checked one sentence by one sentence.

Thank you very much again for your suggestions.

Best regards

Li Liang
